# Enhancing Photoelectric Powder Deposition of Polymers by Charge Control Substances

**DOI:** 10.3390/polym14071332

**Published:** 2022-03-25

**Authors:** Björn Düsenberg, Sebastian-Paul Kopp, Florentin Tischer, Stefan Schrüfer, Stephan Roth, Jochen Schmidt, Michael Schmidt, Dirk W. Schubert, Wolfgang Peukert, Andreas Bück

**Affiliations:** 1Institute of Particle Technology, Friedrich-Alexander-Universität Erlangen-Nürnberg, Cauerstraße 4, D-91058 Erlangen, Germany; bjoern.duesenberg@fau.de (B.D.); florentin.riedel@fau.de (F.T.); jochen.schmidt@fau.de (J.S.); wolfgang.peukert@fau.de (W.P.); 2Collaborative Research Center 814—Additive Manufacturing, Am Weichselgarten 9, D-91058 Erlangen, Germany; s-p.kopp@blz.org (S.-P.K.); s.roth@blz.org (S.R.); michael.schmidt@lpt.uni-erlangen.de (M.S.); 3Bayerisches Laserzentrum Gemeinnützige Forschungsgesellschaft mbH, Konrad-Zuse-Straße 2-6, D-91052 Erlangen, Germany; 4Erlangen Graduate School in Advanced Optical Technologies (SAOT), D-91052 Erlangen, Germany; 5Institute of Polymer Materials (LSP), Friedrich-Alexander-Universität Erlangen-Nürnberg, Martensstraße 7, D-91058 Erlangen, Germany; stefan.schruefer@fau.de (S.S.); dirk.schubert@fau.de (D.W.S.); 6Institute of Photonic Technologies, Friedrich-Alexander-Universität Erlangen-Nürnberg, Konrad-Zuse-Straße 3/5, D-91052 Erlangen, Germany

**Keywords:** additive manufacturing, charge control substances, photoelectric powder deposition, polypropylene, dry coating

## Abstract

Charge control substances (CCS) as additives for polymer powders are investigated to make polymer powders suitable for the electrophotographic powder deposition in powder-based additive manufacturing. The use of CCS unifies the occurring charge of a powder, which is crucial for this novel deposition method. Therefore, commercially available polymer powder is functionalized via dry coating in a shaker mixer with two different CCS and analyzed afterwards. The flowability and the degree of coverage of additives on the surface are used to evaluate the coating process. The thermal properties are analyzed by use of differential scanning calorimetry. Most important, the influence of the CCS on the powder charge is shown by measurements of the electrostatic surface potential at first and the powder deposition itself is performed and analyzed with selected formulations afterwards to show the potential of this method. Finally, tensile strength specimens are produced with the conventional deposition method in order to show the usability of the CCS for current machines.

## 1. Introduction

Powder-based additive manufacturing (laser-powder bed fusion—L-PBF) offers decisive advantages over conventional manufacturing. Examples include the high flexibility in component production, the design freedom in geometries and the lower material consumption [1]. In particular, the use of polymer powders is the focus of current development. The most widely used polymer in L-PBF is currently polyamide 12 with about 90% of the market share [2]. Many other materials such as polybutylene terephthalate [3,4]; polyethylene terephthalate [5]; polyamide 11 [6,7]; polyether ether ketone [8,9]; polystyrene [10]; and polypropylene (PP) [11,12]; as well as polymer composites, for example, polybutylene terephthalate—polycarbonate [13,14,15] are used. However, not all have made it to market maturity yet due to the high demands on the powder properties, such as good flowability and a narrow particle size distribution.

Although the intrinsic properties of the polymer have a fundamental influence on the selection of the material, for example, the melting point, the degree of crystallinity or the mechanical properties [2], the extrinsic properties such as the particle shape, surface roughness and size [16,17] have a similar influence [18]. Specifically, the shape [19], size and inter-particle distance [20] determine the flowability, which is a crucial property for the powder deposition [21,22,23] and, therefore, the processability of the powders. In conventional L-PBF machines, the building chamber is lowered by 100–150 µm before a new powder layer is deposited by a doctor blade or a roller. Afterwards, the powder is molten in predefined areas and the process restarts by lowering the building chamber [2]. Due to an unfavorable particle size distribution, or a non-ideal particle shape, many materials cannot be properly deposited. The ideal particle size for L-PBF is between 70 and 120 µm to match the desired layer thickness [17]. Unfortunately, this size makes the particles significantly more cohesive, which means that the particles have to be functionalized to reduce cohesion and improve flowability [24,25,26]. Another influencing factor in powder deposition, which is often overlooked in this context, is the triboelectric charging of the powders during the deposition. Especially in the heated building chamber, the conditions are ideal for charging the powder layers and, thus, endangering the deposition process.

Triboelectric charging can be observed on many occasions. It often occurs during pneumatic transport, dry particle coating, during fluidized bed processes or is generally spoken in gas phase processes [27,28,29,30]. Triboelectricity depends on many process properties, such as material temperature; gas humidity and pressure; the material itself; the purity and the materials in contact; possible functionalization of the materials; roughness of all surfaces in contact; and the duration, intensity and frequency of contact [31,32,33,34,35,36,37]. Especially a dry process environment, resulting from a high process temperature, leads to higher charges, which result from a low [38] surface humidity on the particles. In an ambient environment (25 °C, 60% humidity), the surface humidity on the particles grounds the powder, which reduces the accumulation of the charge [29,39]. There are various models that describe the transfer of charge by friction. These can be modelled very well for conductive materials [27,40], whereas there are no sufficient models for the charge transfer in insulators such as, for example, polymers [41]. An important element for the understanding of triboelectricity in the sense of this work is that material contacts basically accumulate the charge on the polymer surface. Polymers have a bimodal charge distribution [29] with an integral charge around zero, which means they appear (during charge measurements) almost neutral even when they are highly charged.

A promising new approach for powder deposition in L-PBF is to use photoelectric powder deposition (PED) [42,43,44,45,46] instead of a doctor blade or roller and, therefore, make use of the powder’s ability to acquire the otherwise unwanted charge. Currently, this deposition method is not established in L-PBF; nevertheless, there are efforts to change this [44,47,48]. PED is already the standard deposition process for toner particles in 2D laser printers [49,50]. For the use of PED, the chargeability and the charge control of the material are the most important properties, as the charge has to be either positive or negative to deposit them [49], while the other properties become negligible. The electrophotography consists of the following steps [49,51]: in PED, first, the photoconductive drum is homogeneously charged completely.. Second, the generation of the charge pattern by discharge of the unwanted area using a Laser or LED. Third, there is a transfer of the (in this work) triboelectrically charged powder from the magnetic brush drum to the transfer roll by means of an electric field. Finally, the powder is deposited and fused [49,51]. The largest difference between toner particles and polymer particles for L-PBF is the particle size. Toner particles consist of a polymer, a pigment and at least one charge control agent, as an agglomerate; the particle size distribution is between 6 and 20 µm [23,50,52,53]. The polymer powders used in additive manufacturing usually have a much broader and larger particle size distribution.

To control the powder charge, currently chromium-di-t-butylsalicylic acid and oxycarboxylic acid complexes, zinc-di-t-butylsalicylic acid and oxycarboxylic acid complexes, aluminium-di-t-butylsalicylic acid and oxycarboxylic acid complexes as well as the mixtures of further chemicals of these are added [50,54,55,56,57] to the manufacturing process of the polymers as charge control agents. Since the charge control of polymers is a rather special request, the special products are expensive and not easy to obtain. In newer works, silanized nano titanium dioxide (TiO_2_) was used to control the charge of toner particles as well [58]; the TiO_2_ was deposited via dry mixing in a food processor for a proof-of-concept and came close to a dry coating process.

The most used flowing aid, nano silica, is easily deposited on the polymer powder by dry coating. This is a common way to functionalize powders for L-PBF and has many advantages in comparison with the current production of chargeable polymers. First of all, commercially available powder can be functionalized, as the CCS are applied as guest particles (GP) to the polymer host particles (HP) as Alonso described [59,60]. This means that small and big batches can be produced without much effort. Second, dry coating is a solvent free process, which means that neither a solvent nor a further drying step is required after functionalization as compared to wet coating processes [61,62]. Another benefit, which should not be underestimated, is that the GP serve as flow aids, as they increase the distance between the polymer particles and, therefore, reduce the adhesive forces [20,63,64]. Through these benefits, dry coating is the ideal method to deposit CCS.

Usually, mass fractions between 0.1 wt.% [65] as the lowest amount and 0.5 wt.% (regularly) are necessary [66,67] to functionalize polymer powder sufficiently. Since the flow aids do not melt at comparable temperatures (T_melt,SiO2_ ~1700 °C [68]) as polymers and, therefore, do not fuse within the additive manufacturing process, components with a high silica content are usually very porous and cannot fulfil the desired mechanical properties. For this reason, the lowest possible additive content is desired from the production side. This applies to non-charge control substances as well as for CCS.

In summary, the currently available literature shows that the charge control of polymer particles, especially for the photoelectric powder deposition in L-PBF, is currently only performed via chemical functionalization [50,54,55,56,57]. Some of those products are available on the market, but the use cases are limited at present. However, the charge control by nanoparticulate additives is known and well established from toner technology [49,50]. The functionalization by CCS of the commercially available polymer particles by means of dry coating is therefore the next logical step in the development of this deposition method. In the long term, the subsequent functionalization makes the development of new materials much easier and thus the production process in additive manufacturing more flexible. As an additional benefit, besides the method for material functionalization, an updated state of the art electrophotography in L-PBF is presented, as well as a series of analytical methods to verify functionalization. The aim of this work is, therefore, to use the dry coating for the functionalization of PP with silanized SiO_2_ as CCS and to analyze the influence of the used additives on the powder as well as the deposition methods. Specifically, the photoelectric powder deposition should be enabled. The analysis is performed by means of flowability, the electrostatic surface potential and a thermal analysis via differential scanning calorimetry to determine the degree of crystallinity. Afterwards, the powder deposition of the functionalized powders is analyzed, mechanically by means of a doctor blade and the production of tensile strength specimens, as well as electrophotographically by a model system and the initial development of a powder layer in a building chamber.

## 2. Materials and Methods

For the experiments, commercial polypropylene powder (Coathylene PD0580, Axalta Polymer Powders, Pratteln, Switzerland) with a 10% percentile of the volume sum distribution, x_10,3_ = 33 ± 2.5 µm, the mass-median diameter x_50,3_ = 98.6 ± 3 µm and 90% percentile of the volume sum distribution, x_90,3_ = 186.5 ± 3 µm (determined by the laser diffraction method (Mastersizer 2000, Malvern, UK) is used. The untreated powder is not suitable for the L-PBF process. The density is 0.907 g cm^−3^ (self measurement) and according to the manufacturer [69]. The melting point has its peak at 165 °C, and the crystallization enthalpy is approx. −105.5 J g^−1^.

The polypropylene is either coated with fumed silica particles HDK H05XT (Wacker, Munich, Germany) or HDK H05TA (Wacker, Munich, Germany). The CCS for a negative charge (Silica(−)) are functionalized with hexamethyldisilazane (HMDS) and polydimethylsiloxane (PDMS), which makes the particles hydrophobic and leads to a negative charge. The other CCS (Silica(+)) are functionalized with PMDS as well to make them more hydrophobic and with organic amide or ammonium groups, which leads to a positive charge. Both functionalizations of the silica particles are hydrophobic, which makes the silica particles hydrophobic as well and, therefore, very suitable for the dry coating of hydrophobic polymer particles such as polypropylene [70,71]. Further information on the additive particles is shown in Table 1.

### 2.1. Dry Coating with Charge Control Substances

A shaker mixer (T2f, Willy A. Bachofen AG, Muttenz, Switzerland) at 49 rpm is used for the coating experiments. The 60 g batches are mixed in 500 mL aluminum bottles and the 500 g batches in 1.5 L plastic containers. The following matrix (Table 2) lists the formulations, as well as the coating time of the experiments.

The polymer mass is placed in the mixing containers according to Table 2. To achieve a better deagglomeration of the guest particles, glass mixing aids with a particle size between 1.30 and 1.65 mm (Sigmund Lindner GmbH, Warmensteinach, Germany) with a bulk density of 1.5 g cm^−1^ are added, and the two substances are mixed by gentle, manual shaking in order to avoid stratification in the mixing container. Finally, the quantity of CCS, weighed out on a precision balance, is added and the container closed. Afterwards, 9 samples could be coated simultaneously in the mixer. The ratio between polymer and mixing aids is 1:1. The mixing aids are separated afterwards, by the use of a 1 mm (mesh width) stainless steel sieve.

### 2.2. Scanning Electron Microscopy

The polymer particles have been characterized by scanning electron microscopy (SEM) using a Gemini Ultra 55 (Zeiss) device equipped with a through-the-hole detector. An acceleration voltage of 1 or 2 kV has been applied, according to the best image quality. SEM images are taken at appropriate magnifications.

### 2.3. Powder Flow Characterization

An RST-01.01 (Dr. Dietmar Schulze Schüttgutmesstechnik, Wolfenbüttel, Germany) ring shear tester is used to measure the flow behavior of the functionalized powders. Consolidation stresses of 1300, 2600 and 4600 Pa are applied. The flow function $ff_{C}$, defined by Jenike [72], is the ratio between the consolidation stress $\sigma_{1}$ and the unconfined yield strength $\sigma_{c}$ [72,73].

The tensile strength of a powder bed describes the adhesive forces between two layers of powder. A higher value in tensile strength results from higher interparticular forces between the particle layers. The measurement device is developed by Schweiger et al. and adapted by Meyer, and further information can be found in [74,75].

### 2.4. Differential Scanning Calorimetry

The thermal behavior of the powders is determined by differential scanning calorimetry (DSC) to analyze the influence of the CCS on the crystallization. For this purpose, a Polyma 214 (Netzsch, Germany) is used. The samples (weight: 10 mg ± 0.1 mg) are measured in covered aluminum pans (Concavus Lids (Al), NGB817526, Netzsch, Germany) with dry nitrogen gas purging at 40 mL min^−1^.

Since the sintering window, the area between melting and crystallization peak, is influenced by the addition of nanoscale particles, dynamic measurements are carried out to determine the changes. The program used consists of the following steps: from a starting temperature of 20 °C, the temperature is raised to 200 °C at 10 K min^−1^ and held for 60 s. Subsequently, the temperature is cooled down with 10 K min^−1^ until the initial temperature is reached. 

### 2.5. Electrostatic Surface Potential

To determine the electrostatic surface potential (ESP) of the applied layers, the measurement setup from Hesse et al. [29] is used. For this purpose, an automated film coater, type Coatmaster 510, which is equipped with a doctor blade type Multicator 411 (both Erichsen GmbH & Co. KG, Hemer, Germany) and positioned inside a glove box (SICCO Vitrum, Bohlender GmbH, Grünsfeld, Germany) is used. Restrictions to the resulting powder layer thickness can apply according to the particle sizes of the used material. The deposition speed is set to 10 mm s^−1^. On the traverse of the applicator, an electrostatic voltmeter probe model 1017AE is mounted. The probe is connected to an electrostatic voltmeter: type ISOPROBE model 244A (both Monroe Electronics Inc., Londonville, NY, USA). More information regarding the measurement method is given in Hesse et al. [29]; in this work, 3 depositions are made consecutively per powder to determine the change of ESP while depositing.

### 2.6. Powder Deposition

#### 2.6.1. Photoelectric Powder Deposition

The aim of this test method is to maximize the surface coverage on the transfer roll. Only when this essential step has been taken, can the powder deposition in the building chamber be investigated further. In order to perform the experiments, a triboelectric charging system with adjustable external electric field for powder transfer (Epping, Germany) is used (Figure 1). The transfer roll has a diameter of 255 mm and a width of 119 mm. The field strength of the external electric field used for transferring charged particles is set to +192 kV m^−1^ in case of negatively charged particles and set to −192 kV m^−1^ in case of positively charged particles, which corresponds to a transfer voltage of +770 V and −770 V, respectively. The separation width to achieve the needed field strength is set to 4 mm between the magnetic brush and transfer drum. The procedure for powder charging and transferring is as follows. First, 50 g of the powder mixture, consisting of polymer particles and ferromagnetic carrier particles, is produced by mixing for 30 min in the shaker mixer (T2f, Willy A. Bachofen AG, Switzerland) at 49 rpm and left to rest for 24 h to allow charges, generated during mixing process, to decay. Second, the 50 g mixture is filled into the test setup. Third, the simultaneous start of the triboelectric charging process and activation of the electric transfer field, and at last, the time-resolved analysis of the polymer degree of coverage on the transfer drum using a video analysis (Samsung, South Korea) with an optical resolution of 3840 × 2160 pixel and a scanning rate of 60 fps are performed. The video is analyzed afterwards using ImageJ to obtain the degree of coverage, by binarization at a cut-off threshold of 50%, on the transfer drum.

As a final experiment, first layers are deposited electrophotographically in a building chamber using a prototype deposition system; thus, squares with the functionalized powder are developed with pinpoint accuracy. Since the system is still under construction, with a similar layout as the model setup from Figure 1, these layers are not fused.

#### 2.6.2. Mechanical Powder Deposition and Subsequent Powder Bed Fusion

The determination of the influence of CCS on the mechanical powder deposition is carried out by means of a doctor blade in a commercial laser sintering machine EOS P 385 (Electro Optical Systems, Germany) to produce tensile strength specimens (according to DIN EN ISO 3167—test specimen 1BA). The layer thickness is set to 150 µm and a total number of 15 layers are built with a powder deposition speed of 30 mm s^−1^. The building chamber is preheated to 161.5 °C as the melting temperature of PP is 165 °C and should not be exceeded. To melt the polymer, a CO_2_ laser with a power of ca. 17.5 W (CW) is used, which is about 42% of the total power of the laser. The scan speed of the laser is set to 4500 mm s^−1^ with a hatching distance of 0.3 mm. Due to this method, the influence on the deposition method can be determined, as well as the “printability” of the new functionalized powder.

## 3. Results and Discussion

### 3.1. Degree of Coverage

The analysis of the degree of coverage (DOC) on the polymer powder is performed by MATLAB, by use of Otsu’s method [76]. The change in DOC (see Figure 2) over different additive contents shows that the charge control substances achieve the same DOC during dry coating for one hour. This is also confirmed by the corresponding SEM images (see Figure 3).

As discussed above, the CCS behave similarly during the coating process, which is also shown in the SEM images. This means that it is not necessary to change the dry coating as the functionalization process, which is used for coating with NCC nano silica particles, in order to apply the CCS. This results in a low effort to change to a new additive in order to further improve the powder application in the AM.

### 3.2. Flowability

The measurement by a ring shear cell shows an improvement of the flowability with the addition of guest particles in all formulations (Figure 4). This is not surprising, as more guest particles act as spacers between the host particles. It is interesting to note that the improvement in flowability does not increase linearly with the further addition of guest particles, which often is shown [11] and, therefore, expected up to a maximum flowability. The measured flow factors (ffc) of the formulations are very similar. This probably results from the method itself. During the measurement of the flowability, movement occurs between the particles, which leads to friction. The occurring friction between the particles is similar to the friction occurring in regular powder deposition methods and, although not as strong, the friction used during triboelectric charging in PED. Since the powder is functionalized, the CCS already have an effect, but due to the short measuring time and the low rotational speed of the shear cell, the triboelectric charge occurs as not uniform in the powder bed, which results in different flowabilities.

The powder tensile strength decreases with increasing additive content (Figure 5). The absolute tensile strengths are slightly lower for the Silica(−) formulations than for Silica(+); however, the tensile strengths are around and below 1.0 Pa. This could be due to the sample taken, the test quantity (*n* = 5) or an inhomogeneous distribution of the guest particles on the polymer. The last is rather unlikely, as Figure 2 and Figure 3 have shown that the DOC tends to be higher for the Silica(+) samples. In addition, with regard to the results of the ring shear cell (Figure 4), there is nothing which suggests a difference in the flowability of the two formulations.

### 3.3. Electrostatic Surface Potential

The measurements of the ESP with three successive powder depositions are shown below (Figure 6 and Figure 7). In general, it can be seen for all formulations that the surface potential does not directly depend on the number of applications; something similar was already observed by Hesse et al. [29]. Figure 6 (corresponding peak values in Appendix A) shows the results of the Silica(+) formulations. As described, PP charges negatively in contact with itself and most other materials, which leads to a measurable negative surface potential (Figure 7, corresponding peak values in Appendix A). The addition of Silica(+) GP slightly increases the positive part of the surface potential up to the measured maximum of around 130 V. This proves the effect of the CSS used for functionalization.

Figure 7 shows the Silica(−) formulations, which, in contrast to those discussed above, show no trend in ESP. The reason for this is the triboelectrical charge of the polymer. PP is known [77,78,79] to charge negatively when in contact with most materials and itself. In addition, the effect of the CCS now comes into play, which intensifies and unifies the negative charge up to the maximum, which is defined by the material itself in dependency on the hydrophilicity and the humidity [80]. Measurements with raw PP have shown that the ESP changes randomly in regard to the polarity and intensity.

The ESP already reaches −50 V at an additive content of 0.05 wt.% of Silica(−) nano-particles. Whereas the Silica(+) formulations only reach around 30 V at 0.05 wt.%, as described earlier, the tendency of the PP to charge itself negatively must first be compensated. In the following PED experiment, a clear difference in the degree of coverage can therefore be seen with the selected formulations, especially with an additive content of 0.05 wt.%.

### 3.4. Thermal Analysis

The results of the dynamic DSC measurements are summarized in Table 3 and as a graphic in Appendix A. As expected, the enthalpy of crystallization is slightly reduced compared to the unfunctionalized powder as the nanoparticles act as crystallization nuclei [81,82]. The resulting crystallinity (around 90–95% of the original crystallinity) does not affect the L-PBF process any further, and the use of CCS in this combination can be recommended. The slight decrease in the amount of crystallization enthalpy is due to the silane functionalization from the nanoparticles, which are known for increasing the nucleation activity [83]. Furthermore, a change in the crystallization temperature is observed. Crystallization starts earlier when the additives are added, which results in a smaller sintering window. This is not surprising, as nanoparticles are known to act as crystallization nuclei [84]. It is interesting that the Silica(−) particles seem to result in a slightly stronger change, whereby the difference between the formulations is also dependent on the additive content. A possible reason for this is that the more complex silane molecules on the surface of the Silica(−) particles are more likely to promote crystallization than the simple and much smaller ammonium compounds on the Silica(+) particles. However, this has not yet been confirmed and should therefore be the subject of future research. Since the sintering window of all functionalized PP powders is still large enough (around 20 °C), selected formulations are used for the production of tensile strength specimens as shown and discussed in the next section.

Overall, the change in crystallinity, as well as crystallization temperature, is not unusual for formulations consisting of PP and Silica, as already shown in literature [85].

### 3.5. Powder Deposition

#### 3.5.1. Photoelectric Powder Deposition

The results of the PED are discussed and shown (Figure 8) below. For this purpose, the formulations are used, which have the highest and the lowest additive content, as well as uncoated powder (see Table 2, Exp. 1.2, 5.2, 6.2 and 10.2). This makes it possible to define the framework for the use of CCS. As already discussed, PP charges negatively in contact with itself. The Silica(−) formulations, which enhance the negative charge, reach the maximum coverage on the transfer roll already at very low additive contents of 0.05 wt.%, whereas, a 100% coverage for the Silica(+) formulations is reached only at 1.0 wt.% additive content. This observation is coherent with the measurements of the ESP. At low additive contents, the effect of the Silica(+) particles is not as strong as the Silica(−) particles. In addition to the absolute degree of coverage, the homogeneity of the coverage is another quality characteristic. Here one can see, qualitatively, that the coverage homogeneity varies strongly. It can be seen that some areas have good coverage, while other areas are non-covered. This indicates that, with the minimum additive content of 0.05 wt.%, the distribution of CCS in the powder is not optimal for the PED. More research has to be conducted regarding this topic, but the current measurements indicate that the ESP measurement is a suitable method to determine the chargeability for the electrophotographic powder deposition in L-PBF.

In the following table (Table 4), the corresponding values of the degrees of coverage are listed. The letters in the left corners of the images in Figure 8 connect the values to the images. The electrical field for the deposition of (a–c) has a positive polarity, whereas the electrical field for (d–f) has a negative polarity.

The electrophotographic powder deposition is further implemented in the construction chamber of the machine. For this purpose, the formulations (1.2, 5.2, 6.2 and 10.2; see Table 2) are charged and developed as a square. In Figure 9, the influence of the CCS is clearly shown. An empty building chamber can be seen on the left; here, an attempt is made to develop unfunctionalized PP, which is not possible. In the center, a layer of PP + 0.5 wt.% Silica(+) can be seen. Here, it can be seen that not only is the powder transferred, but also the shape of a square can already be seen. On the far right, the result of the transfer of the formulation PP + 1.0 wt.% Silica(+) is shown. The degree of coverage has clearly increased, and the square shape is recognizable. Based on this proof-of-concept, further work will analyze and investigate the triboelectric powder deposition in more detail.

#### 3.5.2. Mechanical Powder Deposition and Subsequent Powder Bed Fusion

For the powder bed fusion and subsequent analysis of the tensile strength specimens, four formulations are produced in larger batches, as shown in the experimental plan, to produce tensile strength specimens in a commercial PBF machine, using the doctor blade as the application method. It is known from the literature [31,32,33,34] that a high temperature and, thus, low surface humidity on the particles allows them to charge faster and retain their charge longer [86]; therefore, a building chamber is the ideal environment to accumulate charge on particles. Initial investigations have already shown triboelectric effects in the powder bed at an ambient temperature [29]. The results show that the material functionalized with the negative charging CCS could not be deposited, even if the powder bed was lowered by 600 µm.

On the other hand, functionalization with Silica(+) drastically improves the deposition behavior. Thus, it is possible to deposit the polypropylene used here already at a mass fraction of 0.05 wt.% guest particles in the building chamber. As a comparison, the powder deposition of polypropylene with NCC silica usually needs 0.1–0.5 wt.% (according to the literature) to be deposited with a doctor blade. This decreases the amount of additives by a factor of 2–10. The resulting tensile strength specimens can be seen in Figure 10.

#### 3.5.3. Characterization of Tensile Strength Specimens

The produced tensile strength specimens are tested with a material testing machine type Z050 (Zwick & Roell, Ulm, Germany) according to DIN ISO 527-2 with a clamping length of 60 mm, a test speed of 0.5 mm min^−1^ up to 0.25% elongation for the Young’s modulus and a test speed of 1 mm min^−1^ until breakage. The results of the mechanical tests of the tensile strength specimens are shown in Figure 11. As can be seen, the specimens with an additive content of 0.05 wt.% have a higher standard deviation than with a higher additive content. This is to be expected as a uniform distribution of a small amount of additives is much less likely than a large amount of additives. Nevertheless, the Young’s modulus is roughly identical for 0.05 wt.% and 1.0 wt.%. Comparing the stress at break and elongation at break for both formulations, higher stresses and elongations have been achieved at low contents. This is also to be expected, as the GPs are not fused and thus the component is less fragile at lower additive contents, which also means that the amount of additives can directly influence the mechanical properties of the component.

## 4. Conclusions

Commercially available polypropylene was functionalized with CCS via dry coating to unify the polymer charge and make the powder suitable for the electrophotographic powder deposition. Furthermore, their benefits on mechanical powder deposition in additive manufacturing were investigated. By analyzing the degree of coverage, it was possible to determine that the CCS could be deposited on the polymer surface. From the measurements with the ring shear cell, it can be seen that the flowability of the functionalized powders improved drastically, which was also proven by measurements of a powder tensile strength tester. Since the crystallization behavior is of great importance for L-PBF, the influence of CCS was investigated via DSC. The degree of crystallization was just slightly influenced by the nanoparticles and was levelling around 90–95% of the original degree of crystallization, which is similar to the functionalization with NCC known from the literature. The crystallization temperature shifted with the additive content and also with the type of additive, thus reducing the sintering window by only 3 °C, which left a suitable working range. By measuring the electrostatic surface potential, the benefit of the CCS could be confirmed. The surface potential was unified by the additives, which is essential for the photoelectric powder deposition. On this basis, the PED was demonstrated using a model setup as well as a prototype to develop a powder square in an actual building chamber. This could potentially bring new materials into additive manufacturing, as the new deposition method eliminates many of the problems of the currently used doctor blade. At last, the powder was used to produce tensile strength specimens by use of a production machine, equipped with a doctor blade. It was shown that the unified charge resulting from the CCS can be helpful even for the classical deposition method. Nevertheless, this was highly dependent on the type of the CCS.

## Figures and Tables

**Figure 1 polymers-14-01332-f001:**
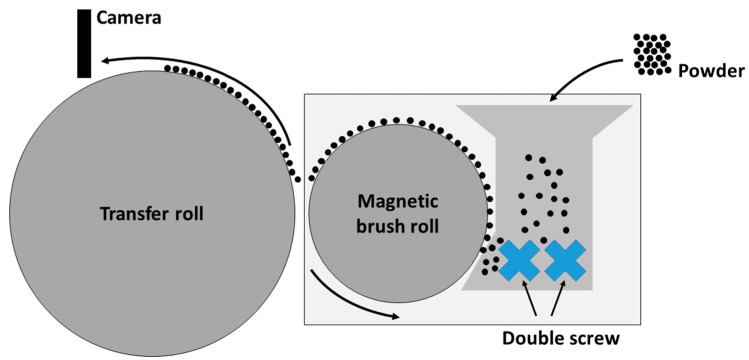
Experimental setup for the analysis of electrophotographic powder deposition (according to [49]).

**Figure 2 polymers-14-01332-f002:**
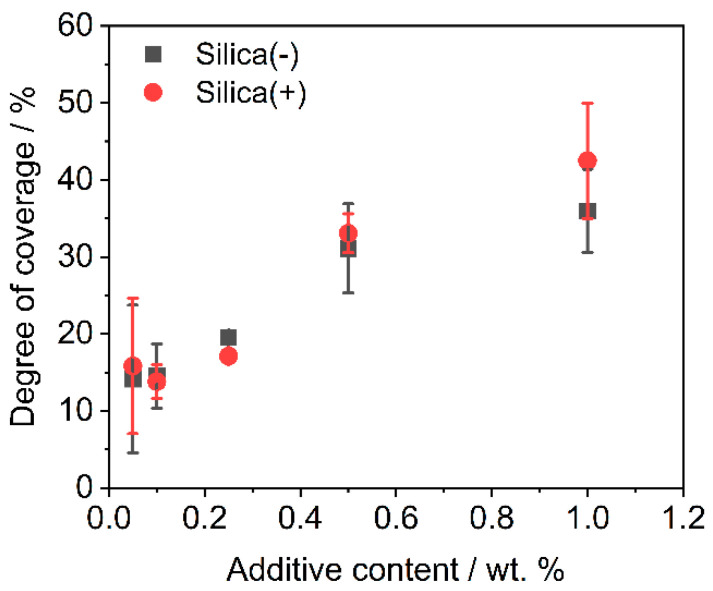
DOC measured after functionalization (*n* = 3) after dry coating for one hour.

**Figure 3 polymers-14-01332-f003:**
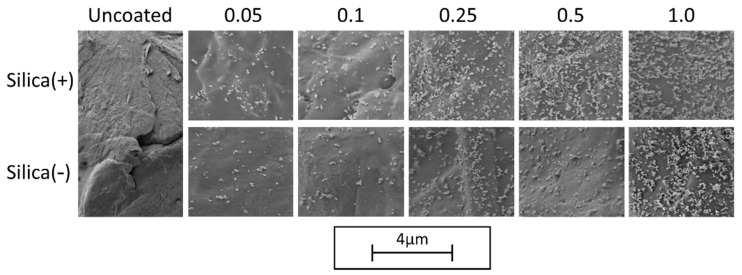
SEM—Images of the DOC with different additive contents.

**Figure 4 polymers-14-01332-f004:**
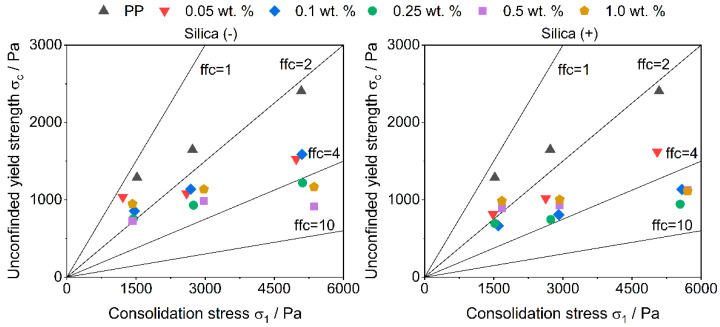
Flowability measured with a ring shear cell.

**Figure 5 polymers-14-01332-f005:**
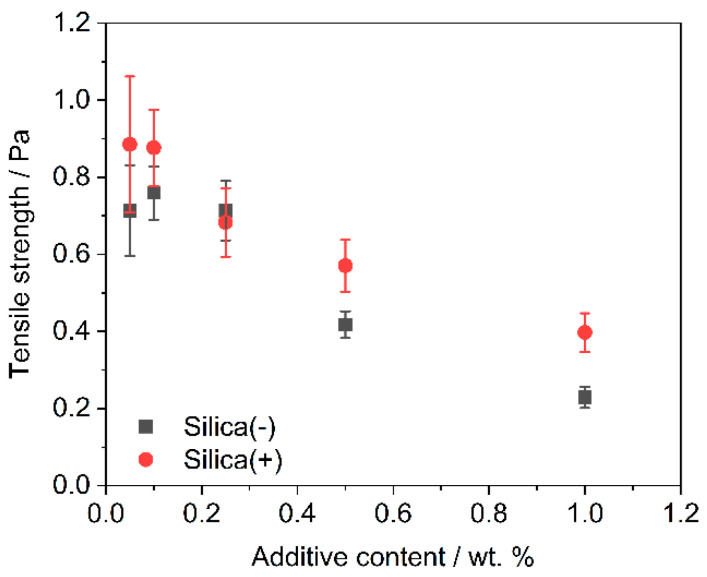
Plot of tensile strength of all formulations grouped by additive content (*n* = 5). Silica(-) in black, Silica(+) in red; the tensile strength of unfunctionalized PP is 4.8 Pa ± 1.2 Pa.

**Figure 6 polymers-14-01332-f006:**
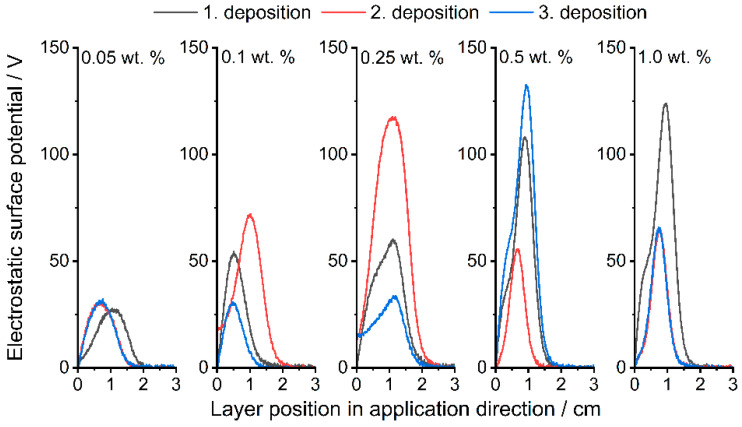
Measured ESP at different layer positions. Comparison between the Silica(+) formulations; increasing content from left to right.

**Figure 7 polymers-14-01332-f007:**
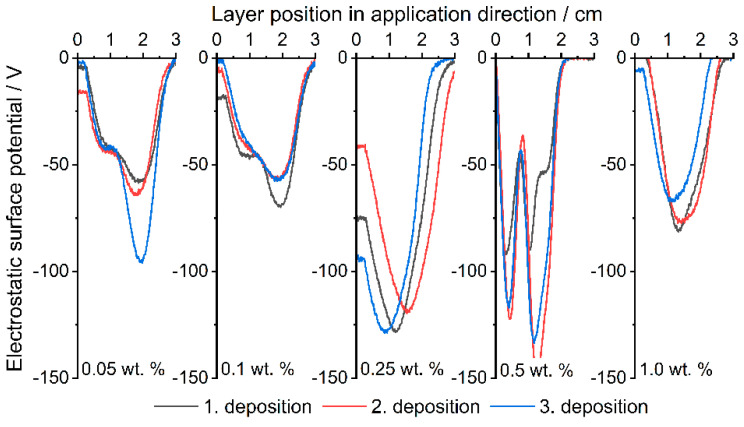
Measured ESP at different layer positions. Comparison between the Silica(−) formulations; increasing content from left to right.

**Figure 8 polymers-14-01332-f008:**
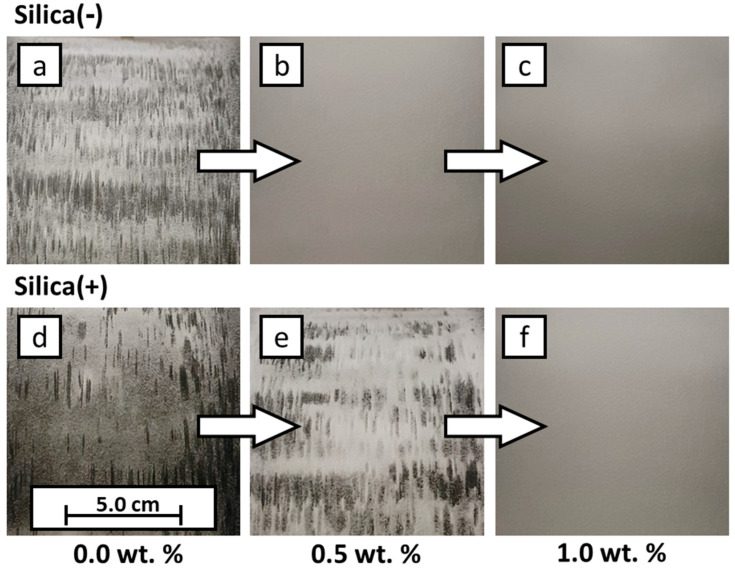
Results of electrophotographic powder deposition. Extracts from the image evaluation showing the powder distribution on the transfer roll. The corresponding degrees of coverage are listed in Table 3. (**a**) raw PP, transfer roll with positive electric field; (**b**) formulation 1.2, transfer roll with positive electric field; (**c**)) formulation 5.2, transfer roll with positive electric field; (**d**) raw PP, transfer roll with negative electric field; (**e**) formulation 6.2, transfer roll with negative electric field; (**f**) formulation 10.2, transfer roll with negative electric field.

**Figure 9 polymers-14-01332-f009:**
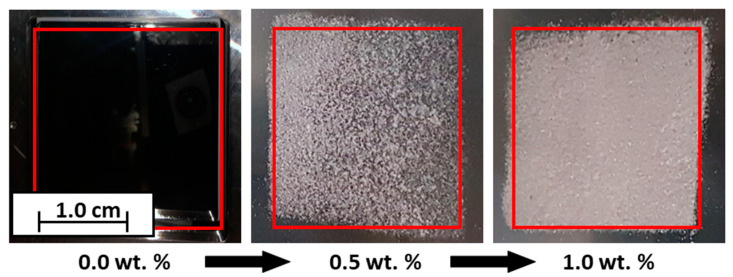
Example results of the electrophotographic powder deposition in a building chamber. From 0.0 wt.% CCS, no powder could be deposited photoelectrically; 0.5 wt.% Silica(+), electrophotographic powder deposition possible; 1.0 wt.% Silica(+), degree of coverage increased.

**Figure 10 polymers-14-01332-f010:**
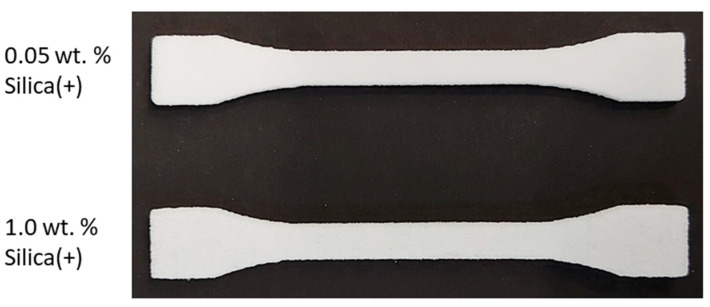
Tensile strength specimens built from PP functionalized with Silica(+).

**Figure 11 polymers-14-01332-f011:**
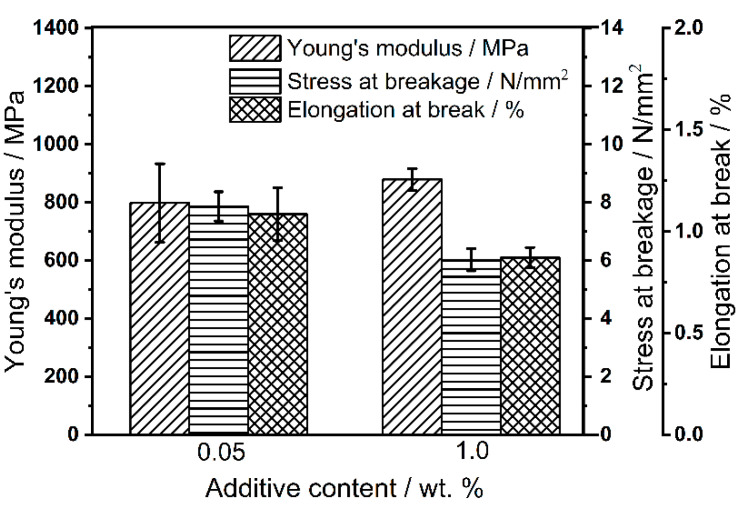
Mechanical properties of the tensile strength specimens from Silica(+) formulations.

**Table 1 polymers-14-01332-t001:** Material properties of HDK H05XT and HDK H05TA according to the manufacturer.

Property	Silica(−) HDK H05TX	Silica(+) HDK H05TA 3
Particles tend to charge	negative	positive
Specific surface area (BET)/m² g^−1^	50 ± 20	50 ± 20
Mean particle size/nm	50	50
Agglomerate particle size/µm	<20	<20
Specific charge/µC g^−1^	−450	+50
Surface modification	HMDS/PDMS	PDMS/−NR_2_/−NR_3_

**Table 2 polymers-14-01332-t002:** Plan of deposition experiments.

No	Polypropylene	Silica(−)	Silica(+)	Mixing Aids	Coating Time
-	g	wt.%	wt.%	g	Min
1.1	3 × 60	0.05	-	60	60
1.2	1 × 500	0.05	-	260	
2	3 × 60	0.1	-	60	
3	3 × 60	0.25	-	60	
4	3 × 60	0.5	-	60	
5.1	3 × 60	1.0	-	60	
5.2	1 × 500	1.0	-	260	
6.1	3 × 60	-	0.05	60	
6.2	1 × 500	-	0.05	260	
7	3 × 60	-	0.1	60	
8	3 × 60	-	0.25	60	
9	3 × 60	-	0.5	60	
10.1	3 × 60	-	1.0	60	
10.2	1 × 500	-	1.0	260	

**Table 3 polymers-14-01332-t003:** Measured values of the degree of the thermal analysis.

Formulation (cf. Table 2)	Enthalpy J g_polymer_^−1^	Crystallization Temperature °C
Raw PP	−106	120
1.1	−96.9 ± 0.6	122.7 ± 0.08
2	−97.4 ± 1.2	122.2 ± 0.001
3	−95.6 ± 1.1	122.4 ± 0.001
4	−96,6 ± 2.0	122.8 ± 0.12
5.1	−99,4 ± 1.8	123.4 ± 0.12
6.1	−94.9 ± 0.9	122.1 ± 0.05
7	−97.6 ± 0.6	122.3 ± 0.05
8	−96.5 ± 1.7	122.3 ± 0.12
9	−96.4 ± 1.3	122.4 ± 0.08
10.1	−100.0 ± 0.6	144.8 ± 0.05

**Table 4 polymers-14-01332-t004:** Measured values of the degree of coverage on the transfer roll.

Corresponding Image from Figure 8	Formulation (cf. Table 2)	Degree of Coverage/%	Polarity of the Electrical Field of the Transfer Roll
a	Raw PP	73.1 ± 4.8	Positive
b	1.2	99.74 ± 0.5	
c	5.2	99.6 ± 0.3	
d	Raw PP	66.14 ± 7.3	Negative
e	6.2	75.26 ± 11.0	
f	10.2	100 ± 0	

## Data Availability

The data that support the findings of this study are available from the corresponding author, upon reasonable request.

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
