# Peer review of "Enhancing Photoelectric Powder Deposition of Polymers by Charge Control Substances"

_polymers, 2022, doi:10.3390/polym14071332_

Round 1

Reviewer 1 Report

The paper is interesting for different groups of researchers working in various fileds such as materials for additive manufacturing, mechanical properties and powder technology. The subject is worthy of investigation and the manuscript has a rich and suitable methodology and is clearly written. With such strong points  the paper deserves publication but before  is a revision  need of following aspects :                                                                                   1 a better presentation of what is really new in the manuscript based on a comparison with the existing data in literature;

      2 a better data organization taking into account that being two tables and eleven figures a part of results from figures have better place in tables. It is to mention as examples values of electrostatic surface potential and of entalpy of crystallization entalpy as a function of additive content which could benefit .

 3  the photoelectric powder deposition  with results presented in figures 9 and 10  could be improved as well, moving a part of data in tables and making the SEM images more clearly with arrows.

4.  the title of table two  is not a plan of all experiments being a plan only for deposition experiments. 

5.  a part of supplementary file has  a better place  in the manuscript 

Reviewer 2 Report

Dear Editor, the aim of this work is to use the dry coating for the functionalization of PP with silanized SiO2 as CCS and to analyze the influence of the used additives on the powder as well as the deposition methods. The paper is well organized and contains new and interesting data. For this reason, I propose to accept it for publication.

Some minor importance comments.

What are the x10,3, etc.? There are no explained in the text.

The experimental procedure for dry Coating with charge control substances, is not well described and I think that if someone else tries to reproduce is, it would be not possible.

Round 2

Reviewer 1 Report

I do believe that the revised manuscript is a better paper and my recommendation now is to be accepted for publication   in journal Polymers as it is.